# Role of c-Src in Carcinogenesis and Drug Resistance

**DOI:** 10.3390/cancers16010032

**Published:** 2023-12-20

**Authors:** Lukmon Raji, Angelina Tetteh, A. R. M. Ruhul Amin

**Affiliations:** Department of Pharmaceutical Sciences, Marshall University School of Pharmacy, Huntington, WV 25755, USA; raji1@marshall.edu (L.R.); tetteh2@marshall.edu (A.T.)

**Keywords:** c-Src, v-Src, carcinogenesis, drug resistance, signal transduction

## Abstract

**Simple Summary:**

The human body relies on essential proteins for cellular growth and development, which are synthesized according to the genetic code contained within the DNA (gene). However, if a gene malfunctions, abnormal proteins can result and lead to uncontrolled cell growth, ultimately contributing to the formation of cancerous tumors. The c-Src gene serves as a prime example of a dysregulated gene that results in the synthesis of abnormal protein that has been associated with the development of numerous types of cancer in humans. c-Src is a crucial player in the pathogenesis of cancer in both humans and other animals. It activates several proteins in cancer development, and aids established cancers in evading chemotherapeutic drugs through various mechanisms, facilitating drug resistance.

**Abstract:**

The aberrant transformation of normal cells into cancer cells, known as carcinogenesis, is a complex process involving numerous genetic and molecular alterations in response to innate and environmental stimuli. The Src family kinases (SFK) are key components of signaling pathways implicated in carcinogenesis, with c-Src and its oncogenic counterpart v-Src often playing a significant role. The discovery of c-Src represents a compelling narrative highlighting groundbreaking discoveries and valuable insights into the molecular mechanisms underlying carcinogenesis. Upon oncogenic activation, c-Src activates multiple downstream signaling pathways, including the PI3K-AKT pathway, the Ras-MAPK pathway, the JAK-STAT3 pathway, and the FAK/Paxillin pathway, which are important for cell proliferation, survival, migration, invasion, metastasis, and drug resistance. In this review, we delve into the discovery of c-Src and v-Src, the structure of c-Src, and the molecular mechanisms that activate c-Src. We also focus on the various signaling pathways that c-Src employs to promote oncogenesis and resistance to chemotherapy drugs as well as molecularly targeted agents.

## 1. Introduction

Carcinogenesis involves abnormal, rapid, and uncontrolled cell growth that results from the excessive stimulation of growth drivers (oncogenes), dysregulation of growth inhibitors (tumor suppressors), or both. For many decades, the cause of cancers remained elusive, and multiple theories were postulated to explain this mystery. The experimental work of Peyton Rous provided a scientific explanation for the origin and transmissibility of cancers [1]. Still, his hypothesis and results were largely ignored by the leading researchers of his day. c-Src is an abbreviated form of the cellular sarcoma protein, a member of the SFK, and a potent facilitator of carcinogenesis. It is a ubiquitously expressed protein and a proto-oncogene of great significance in the SFK family, which includes ten different proteins: Src, Frk, Lck, Lyn, Blk, Hck, Fyn, Yrk, Fgr, and Yes of non-receptor tyrosine kinases (RTK) [2,3,4].

c-Src has been implicated in several human cancers; for example, elevated c-Src activity has been found in human sarcoma tissues, head and neck cancer, lung cancer, breast cancer, and various other carcinomas [5,6,7]. The regulation of multiple carcinogenesis-associated cellular processes such as proliferation, adhesion, metastasis, and invasion are under the control of c-Src. Though it remains inactive during normal cellular homeostasis, it may be momentarily activated during mitosis [8]. Src responds promptly to growth factor signals, triggering interactions with cell surface receptors, such as epidermal growth factor (EGF) receptor (EGFR) [7] or platelet-derived growth factor (PDGF) receptor (PDGFR) [9], propagating a cascade of cellular signaling leading to cellular transformation. The constitutively active (CA) Src, or v-Src, is often described as an oncogene because it stimulates excessive cell growth [10]. Both kinases and phosphatases modulate c-Src activity. The hyperactivity and upregulation of c-Src in several cancers make it a potential drug target in cancer therapy, and the Food and Drug Administration (FDA) has approved several Src inhibitors. Although molecularly targeted drugs, particularly kinase inhibitors, are changing the landscape of cancer therapy, the build-up of resistance to FDA-approved kinase inhibitors is posing a severe challenge [11,12], and c-Src plays a critical role in mediating drug resistance. This review discusses the discovery and structure of c-Src, its role in normal physiology and carcinogenesis, and how c-Src contributes to cancer heterogeneity, cancer stem cells, and drug resistance.

## 2. History of the Discovery of c-Src

Over a century ago, Peyton Rous published an article titled “A Transmissible Avian Neoplasm” based on his findings from a series of experiments on chickens [1]. Rous demonstrated that tumors are transmissible between birds of the same breed using Plymouth Rock chickens in this case. In his initial experiment, Rous inoculated a healthy bird with tumor cells from the bird suffering from an abnormal growth in its breast. The healthy bird developed a sarcoma-like growth in its breast after 35 days [1]. His novel findings would ultimately change the focus of tumor biology. Though Ellerman and Bang had demonstrated that certain leukemias are transmissible by filtrates devoid of host cells two years before Rous published his findings, both studies were initially met with indifference since, at the time, it was thought that leukemias and avian neoplasms were unrelated and non-infective entities [13,14]. In 1911, Rous decided to conduct other fowl experiments identical to the Ellerman and Bang experiment. He inoculated a healthy hen with cell-free filtrates from a fowl with tumor growth in the chest region, and he observed similar results identical to his first experiment [15]. They described the tumor-causing agent as a “transmissible mutagen”, later called Rous Sarcoma Virus (RSV) [15,16].

Although his findings pioneered viral oncogenesis and facilitated the discovery of several mammalian oncogenic viruses [17], his hypothesis and additional experimental results were initially greeted with incredulity because they conflicted with a prevailing concept that the etiology of tumor growth is from an intrinsic causative agent and not a transmittable extrinsic hypothetical virus [18]. Different pathologists described his findings as irrelevant, and one pathologist specifically said: “Young man, if you found its transmissible cause, then it is not cancer”. Rous tried his best to describe the virus adequately, but his words had little impact on the renowned elites of his days [16]. The central point of their argument was that if a single pathogenic agent is responsible for cancer, how does it cause tumor growth in different parts of the human body and diverse parts of the world? Rous attempted to isolate the virus from a mammalian host with little success. He could not convince his colleagues and ultimately moved to other research areas but later returned to cancer research in the 1950s [18]. Although he was unable to define the cellular components (gene/protein) responsible for viral transformation, Peyton Rous was awarded the Nobel Prize in Physiology or Medicine in 1966 at the age of 87, only three years before his passing. He was recognized for his discovery of a “transmissible mutagen”, which was a significant breakthrough [16,18]. After Rous’ experiments, other virologists explored the carcinogenicity of the RSV and developed different strains. These strains, including Bryan-RSV, Prague-RSV, Schmidt–Ruppin-RSV, and Mill–Hill-RSV, have been shown to cause tumors in mammals like rats and rabbits [13].

Hidesaburo Hanafusa, an immigrant from Japan, conducted multiple experiments on chick embryo cells after being inspired by Rous’ experiment [19]. Alongside his wife Teruko, he infected chick embryo cells with a diluted high-titer strain of Bryan-RSV [20]. After thorough observation, they noticed that the normal chick embryo cells had been transformed into cancerous cells, and interestingly, certain foci of the transformed embryo cells did not generate the infectious virus [20]. In their 1963 article, they concluded that the RSV itself is a defective virus incapable of transforming cells in the absence of an infectious helper virus [20]. Additional investigations by Dr. Hanafusa revealed that within the high-titer Bryan-RSV strain, a transformation-defective RSV variant existed that exclusively generates an infectious virus only when a helper virus called Rous-Associated Virus (RAV) is present [21]. A substantial distinction was established between cellular transformation and the production of infectious viruses by transformed cells. These findings paved the way for further explorations into the genetic composition of viruses and furnished crucial clues about the existence of viral oncogenes. Hidesaburo was deeply intrigued by the mystery of RSV transformation and worked with multiple researchers to conduct additional experiments using transformation-defective mutants of the RSV. They were able to map the specific location of the sarcoma gene on the RNA of the Schmidt–Ruppin-RSV strain and named it the *Src* gene [22]. Using the same Schmidt–Ruppin-RSV strain, Brugge and Erikson induced tumors in rabbits. The serum from the infected rabbits was used to immunoprecipitate a 60 kDa protein called pp60^v-Src^ [3]. This protein, pp60^v-Src^, was later referred to by many as viral Src or v-Src. A group of four researchers, namely Michael Bishop, Harold Varmus, Dominique Stehelin, and Peter Vogt, developed hybridization probes that target the *Src* gene [23]. Using these hybridization probes with transformation-defective deletion mutants, they depleted viral sequences unrelated to the transformation agent. This led to the fascinating discovery that the DNA of normal human cells contained the *Src* gene, which they named c-Src [23]. This discovery led to the discovery of many other cellular oncogenes named proto-oncogenes. Michael Bishop and Harold Varmus were awarded the Nobel Prize in Physiology or Medicine in 1989 for discovering the Src gene in humans.

## 3. The Structure of c-Src and the Mechanism of Its Activation

c-Src is a 60 kDa protein encoded by the *Src* gene belonging to a family of non-RTK known as SFK. As a typical protein, the polypeptide sequence of c-Src comprises an N-terminal head and a C-terminal tail but has four Src homology (SH) domains numbered 1, 2, 3, and 4 (Figure 1) [24,25]. The SH1 domain is the active enzymatic kinase segment that bears a swinging amino acid tyrosine—Tyr 416 in chicken Src (Tyr 419 in humans). A linker region next to this domain is followed by the SH2 and SH3 domains. The SH2 and SH3 domains function as an on/off switch for the Src in conjunction with the C-terminal tail. The last domain is the N-terminal SH4 domain; this domain is the leading site of co-translational modification by myristoylation (c-Src and Blk) and post-translational change (other members of SFKs) by palmitoylation [4,26,27]. The C-terminal tail consists of a carboxy group that is short and connected to the SH1 domain. Additionally, the C-terminal domain serves as an anchor for a phosphotyrosine moiety, specifically Tyr530 in human Src (or Tyr527 in chicken). This allows it to function as a regulatory unit. Figure 2 shows how the basal activity of c-Src is maintained during normal physiological conditions and the mechanism of its activation. The presence of an intrinsically disordered segment, known as the unique domain (UD), is responsible for the variation among SFKs. This domain lies between the SH4 and the SH3 polypeptide domain [26]. The structure of c-Src is a fuzzy complex of different SH1, 2, 3, and C-terminal domains, which are coiled up into a twist. At the same time, the SH4 and the UD are somewhat intrinsically disordered [28].

### 3.1. SH1 Domain

The SH1 domain is the catalytic arm of the c-Src, encoding a cAMP-dependent protein kinase [29]. The SH1 domain consists of two lobes: a small N-terminal lobe and a large C-terminal lobe. These lobes are connected by an elastic bridge that acts as a binding pocket for ATP as well as for inhibitors (Figure 1). The C-terminal region of the SH1 domain comprises an α-helix, whereas the N-terminal region consists of numerous filaments of antiparallel β-sheets, with a connector between the β3 and β4 strands [30,31,32]. This C-terminal lobe contains multiple amino acids, including Tyr 416 (chickens) or 419 (humans), responsible for the catalytic abilities of c-Src and substrate phosphorylation. Dephosphorylation of Tyr530 by various phosphatases and subsequent autophosphorylation of Tyr419 facilitates the phosphorylation of various substrates that regulate cellular migration, adaptation, mitogenesis, and angiogenesis [33,34,35,36]. A kinase-dead mutant, K295M, generated less phosphorylated tyrosine than CA-Src (Y527F), which is suggestive of the SH1 domain being crucial in Src-induced carcinogenesis [37].

### 3.2. SH2 Domain

The SH2 domain is a crucial segment of c-Src because it binds the phosphotyrosine Tyr530 on the carboxy-terminal tail (Figure 2) and also serves as an attachment point for c-Src substrates [38]. The architecture of this domain is composed of a network of approximately 100 residues of amino acids, which are interwoven into α-helices and β-sheets. The β-sheets are made up of four filaments of amino acids, part of which includes a cysteine residue, CysβC3 that binds phosphotyrosine with low affinity on the carboxy tail in a lock and key mechanism; this helps c-Src achieve an inactive conformation [31,39,40]. The SH2 domain specifically binds to phosphorylated tyrosine residues. However, in v-Src, the lack of the carboxy tail causes a loss of interaction between the CysβC3 and phosphotyrosine. This leads to the recognition of v-Src as a CA-Src since it no longer has its regulatory carboxy tail that binds the SH2 domain. v-Src has ten cysteine residues that can undergo single or multiple mutations. These residues play a crucial role in the deactivation of v-Src by SH-alkylating agents. Notably, the SH2 domain contains three of these cysteine residues, namely Cys185, Cys238, and Cys245 [41].

The α-helices of the SH2 domain are loosely packed and serve as intermediary points between the SH2 domain and SH1 domain; the surfaces of these domains have several water molecules attached with electrostatic interactions [31]. The altered SH2 domain is paramount to the carcinogenicity of the Src kinase. This domain helps c-Src associate with partner molecules such as phosphatidylinositol-4,5-biphosphate 3-kinase (PI3K) [42], Grb2, Janus kinase (JAK), SHP1/2, Cbl, and non-catalytic region of tyrosine kinase (NCK) [43,44,45,46,47]. The R175A mutant, which has its SH2 domain inactivated, scarcely interacts with c-Src substrates [37]. In addition, when treated with kinase inhibitors in a drug resistance study, the R175A mutant did not localize rapidly at the focal adhesions compared to the wild-type c-Src [36].

### 3.3. SH3 Domain

The SH3 domain consists of a fold of five β-strands of approximately 60 amino acid residues (Figure 3) with three hydrophobic cavities that act as the ligand-binding domain [48]. Anatomically, the ligand-binding motif of the SH3 domain binds proline-rich sequences, especially those with PxxP motifs. The PxxP motif is found in class I (those with R/KxXPxXP sequence) and II peptides (those with XPxXPxR/K sequence) where “X” represents non-glycine hydrophobic residues and “x” represents natural amino acid residues [48]. These residues are often orientated to form a left-handed polyproline helix that exhibits pseudo-syllogism [49]. The SH3 domain recognizes and binds the proline-rich motifs; these interactions help to further stabilize the closed conformation and maintain c-Src in a folded inactive state [50]. The molecular structure analysis of the SH3 domain of c-Src by X-ray crystallography and nuclear magnetic resonance (NMR) shows three main loops that allow the five β-strands of the SH3 domain to maintain their folds [51]. The Src kinase SH3 domain contains three loops: the RT Src loop, the n-Src loop, and the distal loop. The RT Src loop is composed of two main amino acid residues, arginine (R) and threonine (T) [52]. The RT loop also interconnects with the β1 and β2 strands, the n-Src loop interconnects the β2 and β3 strands, and the distal loop interconnects the β3 and β4 strands [53]. These loops also form hydrogen bonds and van der Waals forces with the SH1 domain [31]. The RT (residues 98–103) and n-Src (residues 114–116) loops of the SH3 domain contain a lipid-binding region positioned antithetically to the hydrophobic peptide cavities. This lipid-binding region has the potential to bind the SH4 domain [54]. A W118A mutant (Src-Y527/W118A) with mutations in the SH3 domain did not abolish the interaction of Src with partner molecules. This suggests that this domain is not essential to halting the kinase activity of c-Src [37]. The SH3 domain is next to the SH2 domain, and the hydrophobic amino acid residues on these domains interact via a helical twist [31].

### 3.4. The N-Terminal SH4 and the Unique Domain

The glycine residue containing the N-terminal SH4 domain is the primary site for membrane modification. The Src protein undergoes lipid membrane changes only through myristoylation, which is essential for its association with the cell membrane. This process involves the covalent transfer of myristic acid by the enzyme myristoyl-coenzyme A to the N-terminal glycine residue of c-Src [55]. In addition to being associated with the membrane, the myristoylated N-terminal of c-Src also interacts with the SH3 domain, even in the absence of membrane lipids, forming a mesh [56]. A c-Src G2A mutant that cannot undergo myristoylation at the N-terminal is incapable of causing cellular transformation. This mutant is mainly found in the cytoplasm and does not bind to the membrane [57]. Membrane interaction is pivotal to activating c-Src by phosphatases in the lipid bilayer. In an in vitro drug resistance study in *Xenopus laevis* XTC cells, the G2A mutant was found not to impair specific kinase inhibitors’ activation of c-Src [36,56].

The SH4 and the UD are often described as a region of intrinsically scattered proteins with low Gibbs free energy; this makes them interact with other proteins and macromolecular complexes [58,59]. The primary function of the UD was a mystery for many years. However, Perez et al. demonstrated that the UD binds acidic lipids and phosphoinositides through its unique lipid-binding region. This interaction is controlled by the phosphorylation of amino acid residues (T37, S75, and S17) in the UD [60]. The binding of acidic lipids to the UD triggers the displacement of the protein core from the cell membrane. This results in a reduction in the protein’s interaction with membrane lipids, which is referred to as “positional regulation”. High levels of Calcium-Calmodulin are responsible for downregulating the binding of the UD to lipids [54]. The UD plays a crucial role in carcinogenesis as it interacts with phosphoinositides, which control important cellular functions such as metastasis, cell growth, migration, differentiation, and metabolism [61]. The regulation of phosphatidylinositol-4-kinase, an enzyme that produces phosphoinositides, is poorly understood. This enzyme is overexpressed in human hepatocellular carcinoma (HCC), which suggests its involvement in the PI3K-AKT pathway. As a result, phosphoinositides are considered potential targets for cancer chemotherapy [62].

## 4. Normal Physiological Role of c-Src

As a ubiquitously expressed protein, c-Src plays a critical role in normal physiology and is important for development. In normal cells, it regulates physiological processes such as differentiation, proliferation, mitogenesis, adhesion, migration, angiogenesis, survival, etc. Using human embryonic stem cell lines H1, H7, and H9, Zhang et al. reported a modest increase in the expression of c-Src level as a function of differentiation [63]. Furthermore, loss of c-Src expression in mice impairs the development of various reproductive organs, including ductal, uterine, and ovarian development [64]. In contrast, deletion of c-Src in mice impaired osteoclast functions, leading to reduced bone remodeling and osteoporosis [65], while deletion in osteoblasts promotes their differentiation and formation of bones [66]. c-Src also promotes lens epithelial cell proliferation in lens development [67].

In normal physiology, c-Src is also activated by various mitogenic growth factors such as EGF, PDGF, and integrins. Chang et al. reported that c-Src is required for EGF-induced mitogenesis in murine fibroblasts [68]. They further demonstrated that c-Src regulates EGF-induced actin cytoskeleton rearrangement through phosphorylation of p190. Overexpressing various SH3 domain mutants of c-Src (Y133F and Y138F) in murine fibroblast cell lines, Broome and Hunter reported that the c-Src SH3 domain is required for EGF- and PDGF-induced mitogenesis [69]. The requirement of c-Src and the SH2 domain was also demonstrated in PDGFR mitogenic signaling in NIH3T3 fibroblasts [70].

Angiogenesis is another normal physiological process that involves the proliferation of endothelial cells (EC). Vascular endothelial growth factor (VEGF) serves as a mitogen for EC. Multiple studies demonstrated the critical role of c-Src in this process [71,72,73,74,75]. Mice deficient in c-Src showed no VEGF-induced vascular permeability, suggesting that VEGF-mediated angiogenesis, particularly vascular permeability, required Src activity [72]. Furthermore, VEGF recruited c-Src to the VEGF receptor-2 (VEGFR-2) in EC, which subsequently activated AKT and Erk1/2 signaling and EC migration, proliferation, and tube formation [71]. Kim et al. demonstrated that TNF-related activation-induced cytokine (TRANCE) induces angiogenesis of human EC through the activation of Src without affecting VEGF expression [76].

Cell adhesion and migration are regulated by cytoskeletal reorganization. Integrin signaling, of which c-Src is a major component, is critical for these cellular functions. Sprouty4 (Spry4), an RTK modulator, regulates EC migration via modulating integrin β3 through c-Src [77]. EC adhesion and migration on vitronectin were inhibited by overexpression and enhanced by the knockdown of Spry4 in a Src-dependent manner. c-Src is required for the expression and tyrosine phosphorylation of integrin β3. c-Src also plays an important role in α5β1- and α4β1-mediated migration of human fibroblasts [78]. Focal adhesion kinase (FAK) may or may not be important for this effect and depends on the specific integrin. Furthermore, the spreading and migration of mouse primary bone marrow-derived megakaryocytes on fibronectin requires Src-Syk signaling since Src inhibitor PP1 and Syk inhibitor R406 abolished these functions [79]. During normal physiological functions, activated c-Src is quickly inactivated by phosphatases.

## 5. Activation of c-Src in c-Src-Dependent Cancer

In most eukaryotic cells, c-Src is dormant (Figure 2) and needs to be activated to interact with partner molecules on the lipid membranes and adaptor proteins to regulate a variety of subcellular localizations and cellular functions [80]; this is achieved by SH2 and SH3 domain repositioning, electrostatic realignment of the salt bridge, dephosphorylation of Tyr530, and an adjustment of the interfering Leu407 in the active segment so phosphorylation can occur at Tyr419. The phosphorylated Tyr419 is highly catalytic and phosphorylates tyrosine residues of other proteins with the aid of two Mg (2+) ions [31,81]. In Src-induced cellular transformation, active Src kinase phosphorylates the tyrosine residues of partner molecules involved in several intercellular signaling pathways. These pathways include the JAK- Signal transducer and activator of transcription (STAT)3 pathway, the Ras-mitogen-activated protein kinase (MAPK)/extracellular signal-regulated kinase (ERK) pathway, the PI3K/AKT pathway, and the FAK/Paxillin pathway.

### 5.1. Src-JAK-STAT3 Pathway

Tumor growth, immunomodulation, and immune-mediated tumor destruction are governed by the JAK-STAT pathways. JAKs are a family of non-RTK consisting of JAK1,2 and 3 and tyrosine kinase 2 (TYK2) that mediate several physiological functions by stimulating type I and type II cytokine receptors. These TKs (JAK1/2 and TYK2) phosphorylate the tyrosine residues of cytokine receptors, which trigger the recruitment and subsequent phosphorylation of STAT proteins [82,83,84,85]. Phosphorylation of STAT proteins can be achieved by several intracellular proteins, including Src kinase [86]. Src-induced STAT phosphorylation leads to STAT dimerization, localization to the nucleus, and consequent gene transcription. Of the different STAT proteins, Src commonly interacts with STAT3 (Figure 4) to promote tumorigenesis, though Src may also phosphorylate other STAT proteins [85]. In recent studies, persistent activation of the progesterone receptor-induced c-Src-JAK-STAT pathway was observed in C4HD breast cancer cells [87]. 

The novel oncogene with kinase domain (NOK) forms a complex with c-Src and activates STAT3 in HEK293 and HCT116 cells, leading to cell proliferation and tumor growth in nude mice [93]. EGFR, which is overexpressed in esophageal, lung, and head and neck cancers, induced metastasis by Src-STAT signaling [5]. Acetylation of c-Src at both C-terminal and N-terminal domains activates STAT3 in NIH3T3, HEK293T, and MCF-7 cells and induces proliferation [94]. c-Src also activates both STAT1 and STAT3 in PDGF-stimulated NIH3T3 cells [95]. In Hela cells. Pyk2, a member of the FAK family, facilitates STAT3 activation through c-Src [96]

### 5.2. Src-Ras-MAPK/ERK Pathway

This pathway is driven by the RAS family of G-proteins, which are small molecular weight GTPases that function as an on/off switch of several molecules involved in cellular homeostasis and activating oncogenic mutations in different human cancers [97]. The pathway is initiated by upstream signals, that cause RTKs such as EGFR to interact with adapter proteins—Shc and Grb2. Shc often complexes with Grb2 and the guanine nucleotide exchange factor—Son of Sevenless (SOS). SOS activates RAS proteins by catalyzing the release of inactive GDP-bound (Guanine diphosphate) RAS proteins, facilitating the binding of active GTP (Guanine triphosphate) [98,99,100,101,102]. Activated RAS undergoes structural rearrangements in the switch I and II regions, promoting interactions with the RAF family of kinases; RAF kinases phosphorylate the resultant product of MAP2K1 and MAP2K2—MEK1 and 2, and the latter phosphorylates cellular effector kinases—ERK1 and 2 (Figure 4) [98].

The MAPK/ERK pathway is turned off by GAPs (GTPase-activating proteins) that hydrolyze active GTP-bound RAS to its inactive GDP-bound state. However, mutations in Gly12, Gly13, and Gln61 in GAP result in defective GAPs; these mutations allow RAS proteins to be permanently on/constitutively active. This is very common in different human cancers; RAS mutants are drivers of cellular proliferation and activators of uncontrolled mitosis [100,103,104].

The relationship between Src and RAS is two-sided; both proteins can activate each other to promote tumorigenesis. Mutated RAS proteins may rapidly facilitate the activation of c-Src in the endoplasmic reticulum and Golgi apparatus; this was found in HEK293 cell lines [105]. v-Src utilizes the Ras-MAPK pathway to facilitate oncogenic cell growth by suppressing the expression of Src homology 2 domain-containing protein tyrosine phosphatase substrate 1 (SHPS-1) in v-Src transformed rat fibroblasts 3Y1 [106,107]. v-Src-mediated transformation of mouse (NIH3T3) and rat (3Y1) suppresses the mRNA and protein expression of focal adhesion protein vinexin by activating the ERK and mammalian target of rapamycin (mTOR) pathways [108]. In the NIH3T3 cell line, Src kinase phosphorylates the Shc adaptor protein, leading to the activation of RAS [109]. At the same time, the mutualistic relationship between Src and RAS accelerates the development of pancreatic ductal adenocarcinoma. Multiple reports suggest that the carcinogenicity of Src kinase is primarily dependent on RAS proteins.

### 5.3. Src-PI3K-AKT-mTOR Pathway

The PI3K-AKT-mTOR pathway is one of the most well-known signal transduction pathways in regulating and facilitating the development of multiple human malignancies, such as breast, HCC, brain, lung, head and neck, prostate, and gastric cancers [110,111,112,113,114,115,116,117]. PI3K is a class of cell membrane-associated enzymes that are master regulators of the cell cycle processes. They are categorized into three major classes, Class I, II, and III, and have three subunits: two regulatory subunits (p85 and p55) and a catalytic subunit (p110). The three subunits form an inactive dimer under normal cellular conditions; the catalytic subunit becomes active when stimulated by hormones, cytokines, and growth factors [118,119,120,121]. The active kinase catalyzes the phosphorylation of several cell membrane phospholipids such as phosphatidylinositol (PI), phosphatidylinositol 4-phosphate (PIP), or phosphatidylinositol 4,5-bisphosphate (PIP_2_). This leads to the activation and recruitment of different downstream proteins and kinases, such as protein kinase B, also known as AKT (A serine/threonine protein kinase) [118,122]. AKT binds to cell membrane phospholipids and facilitates rapid cell growth by the activation of the mTOR and inhibiting tumor suppressor genes. The phosphatase, phosphatase and tensin homolog (PTEN), a negative metabolic regulator, keep this pathway in check by dephosphorylating phosphatidylinositol (3,4,5)-trisphosphate (PIP3) at the D3 position to the biphosphate PIP_2_ [120,123]. This halts multiple partner molecules’ phosphorylation cascade, leading to growth suppression. 

Activation of the PI3K-AKT-mTOR pathway is what Src kinase does the best; estrogen-induced Src activation leads to the phosphorylation and activation of the modulator of the non-genomic action of estrogen receptor (MNAR) in MCF-7 cells [124]. The activated MNAR interacts with the p85 regulatory subunit of PI3K, leading to its activation and subsequent activation of AKT. Cell membrane-bound lipid rafts trigger the interaction of Src with regulatory and catalytic subunits of PI3K, activating the kinase and facilitating its interaction with partner molecules to promote tumorigenesis in a panel of eight small cell lung cancer cells [125]. Additionally, the underlying mechanism by which the hepatitis B virus induces carcinogenesis in hepatoma cells (Huh7 and SK-Hep1) is the Src-PI3K-AKT pathway [126]. Furthermore, FAK promotes integrin-mediated colon cancer cell (SW620) adhesion by the Src-dependent PI3K-AKT-mTOR pathway [127].

### 5.4. Src/FAK/Paxillin Pathway

As the name implies, FAK regulates cell adhesion and cell motility by localization to focal contacts, which are sites enriched with integrins. FAK is a non-receptor tyrosine kinase that helps maintain the cell membrane’s shape by interacting with actin [128,129]. FAK has an N-terminal head four-point-one, ezrin, radixin, and moesin homology (FERM) domain, a central kinase region, a proline-rich domain, and a C-terminal tail which bears the focal adhesion targeting domain [130]. The kinase is activated by lipid binding of the FERM domain and aggregation of focal adhesions to the receptor-bound kinase; this leads to a structural shift causing FAK oligomerization and subsequent autophosphorylation at Tyr397. This autophosphorylation promotes the binding of Src kinase to the phosphorylated tyrosine Tyr397, followed by Src-induced phosphorylation of the tyrosine phospho-acceptor sites—Tyr576 and Tyr577 on the activation loop. This cascade of events generates a fully active Src-FAK complex that can associate with downstream target molecules. The focal adhesion targeting (FAT) domain serves as a binding hub for the association of focal adhesions such as paxillin and talin, which interact with the actin cytoskeleton and integrins, inducing a downstream survival pathway that results in cancer metastasis [128,129,130,131,132,133].

The Src-FAK-Paxillin pathway is responsible for poor prognosis in neuroblastoma patients [134]. Src, FAK, and paxillin proteins are expressed in human neuroblastoma cell lines, SK-N-AS and SK-N-BE, and children with concomitant positivity tumors have poor survival. The collagen type IV alpha 1 chain (COL4A1) is the most significantly overexpressed collagen in HCC and facilitates tumor aggressiveness of HCC cell lines (HepG2, PLC/PRF/5, Hep3B, and SK-Hep1) by activating FAK-Src signaling [135]. This pathway is also activated by Nitrosamine 4-(methylnitrosamino)-1-(3-pyridyl)-1-butanone (NNK) [136], platelet-activating factor (PAF) [137], periostin [138], and leptin [139] in lung (H69, H82, H157, and H1299 cells), ovarian (OVCA429), colorectal (CMT93 and DLD1 cells), and breast (MCF7 and MDA-MB-231) cancer, respectively.

## 6. Role of c-Src in Drug Resistance

One of the most challenging issues facing today’s oncologists in successfully treating their patients is to combat drug resistance (both intrinsic and acquired). Understanding the molecular mechanism underlying drug resistance is crucial for overcoming it. Overexpression of c-Src is a well-known driver mediating drug resistance in various types of cancer, such as breast, ovarian, colon, lung, and head and neck cancer [116,140,141,142,143]. This phenomenon occurs due to c-Src-dependent activation of downstream signaling pathways that promote tumor growth and survival, reducing the efficacy of first-line anticancer drugs. Tamoxifen resistance is a frequent experience for breast cancer patients. Wu et al. reported that an increase in the expression and activation of c-Src leading to activation of the PI3K-AKT pathway in T47D cells causes epithelial-mesenchymal transition (EMT)-mediated tamoxifen resistance [143]. Consequently, the efficacy of tamoxifen, a selective estrogen receptor modulator (SERM), is compromised. However, inhibiting Src with dasatinib restored sensitivity to tamoxifen [143]. Besides mediating tamoxifen resistance, c-Src also drives epirubicin resistance in multidrug-resistant human luminal-type breast cancer MCF-7 cells (MCF-7/ADR) and in MDA-MB-468/EPR cells. This occurred through the phosphorylation of Annexin A2 (Anxa2), a protein implicated in drug resistance [140]. Additional studies in MCF-7/ADR cells by Fan et al. highlighted that Src induces epirubicin resistance in breast cancers by modulating the activity of the drug efflux pump p-glycoprotein (PgP), leading to increased expulsion of epirubicin from the cell [144]. In this case, Rack1 acts as a signaling hub facilitating the interaction of c-Src to PgP, leading to the phosphorylation of caveolin 1 by the Src kinase.

Thymidylate synthase is a central enzyme in the process of DNA synthesis. It plays a critical role in converting deoxyuridine monophosphate (dUMP) to deoxythymidine monophosphate (dTMP), which is required for the synthesis of DNA [145]. 5-fluorouracil (5-FU) is a widely used chemotherapeutic drug that targets the thymidylate synthase pathway, leading to a reduction in dTMP levels and the suppression of DNA synthesis. Src expression led to an overwhelming increase in the expression of thymidylate synthase, which led to a loss of inhibitory activity of 5-FU and resistance to the drug in HCT116 colorectal cancer cells [141].

Cisplatin is among the most widely used chemotherapy drugs. c-Src plays a major role in cisplatin resistance. Peterson-Roth et al. reported that Src can transfer resistance to neighboring cells, in this case, mouse embryonic fibroblasts (MEFs) [146]. This process is facilitated by the phosphorylation of connexin 43, a gap junction protein. The overexpression of Src in a group of MEFs typically hinders the distribution of death signals. In addition, the proto-oncogene E26 transformation-specific sequence-1 (ETS-1), which is frequently overexpressed in many cancers, is upregulated by Src [116]. The overexpression of ETS-1 in cisplatin-resistant head and neck cancer cell lines (Cal27 and Fadu) led to reduced apoptosis and promoted rapid cellular invasion and survival of these head and neck cancer cells. Interestingly, treating the resistant cells with a MEK inhibitor did not reverse their sensitivity to cisplatin. However, when dasatinib was used to inhibit Src, it reduced the expression of ETS-1, and subsequently, the cells underwent apoptosis. Finally, the oncogenic form of Src (v-Src) causes cisplatin resistance by reversing cisplatin-mediated DNA damage in human gallbladder adenocarcinoma cells [147]. v-Src transfected HAG/src3-1 cells showed 3.5-fold resistance to cisplatin but not to doxorubicin, 5-FU, or etoposide. Src promotes DNA repair via a mechanism that is distinct from either the Ras-MAPK, PI3K-AKT, or PKC pathway.

Src is the primary regulator of fulvestrant resistance in estrogen receptor-positive ovarian cancer or T47D breast cancer cells [148,149]. Treatment with saracatinib—a Src kinase inhibitor-induced cell cycle arrest and reduced survival of fulvestrant resistance ovarian cancer cell lines (PEO1 and PEO1R) [142]. More than 70% of colon cancers have enhanced Src activation, which has been shown to facilitate drug resistance in colorectal cancers [150,151,152]. Due to acquired resistance, the FDA-approved first-line medication in colon cancer—5-FU, lacked adequate antitumor activity in the HCT116 colon cancer cell line. Src facilitated this resistance via the upregulation of thymidylate synthase [141]. Moreover, the activation of Src has been linked to resistance to oxaliplatin in six different colon cancer cell lines (HT29, LS174T, SW480, HCT116, KM12-L4, and DiFi) [153]. These cells regained sensitivity to oxaliplatin when Src was inhibited with dasatinib.

In five different lung cancer cell lines (A549, H226, H2009, H1299, and H1792) and xenograft models, c-Src plays a significant role in disease progression and is associated with poor therapeutic outcomes [154]. Resistance to cisplatin, a commonly used drug in the pharmacotherapy of lung cancer, is frequently observed. A549/DDP lung cancer cells that have developed chemoresistance to cisplatin do not often undergo apoptosis [155]. However, when sunitinib inhibited Src, it led to apoptosis in these resistant lung cancer cells. Additionally, the survival of the PC9 lung cancer cell line that has developed resistance to erlotinib through Src-mediated mechanisms is common. Nevertheless, when Src is inactivated, it results in the restoration of sensitivity to erlotinib [156].

Another mechanism behind the resistance induced by Src is the activation of c-MET, also known as the mesenchymal-epithelial transition factor. In head and neck cancer cell lines (MDA686LN, JHU022), resistance to EGFR inhibitor erlotinib was observed, and this resistance appeared to be propagated by the Src-dependent activation of the MET signaling pathway [157,158]. Furthermore, the in vivo xenograft model shows that c-Src activation contributed to erlotinib resistance. Src kinase can also trigger the activation of the drug efflux transporter, PgP, a major cause of multidrug resistance [159]. Activating the drug efflux pump PgP increases the efflux of anticancer drugs, leading to decreased bioavailability. This results in a dampened overall killing effect of Src kinase inhibitors and cell survival.

## 7. Role of c-Src in Tumor Heterogeneity and Cancer Stem Cell

Cancer is a progressive, clonal disease in which normal cells progress to hyperplasia, dysplasia (mild, moderate, and severe), carcinoma in situ, invasive cancer, and finally to metastatic disease. During this lengthy course of progression, cancer cells become heterogeneous. In addition to interpatient and intertumoral heterogeneity, there also exists intratumoral heterogeneity. Each tumor mass consists of a diverse collection of cells with distinct subpopulations that differ in their phenotypic and phenotypic characteristics, including morphology, gene expression, metabolism, motility, proliferation, metastatic potential, and drug response [160]. Within a heterogeneous tumor, there exists a subpopulation of cancer cells that possess the characteristics of stem cells (self-renewal ability) known as cancer stem cells. Tumor heterogeneity and the existence of cancer stem cells pose a significant challenge to the success of tumor treatment. Genetic diversity is the driving force for tumor heterogeneity and cancer stem cells. c-Src is a critical contributor to both tumor heterogeneity and cancer stem cells. To address the role of c-Src, cancer stem cells were isolated from the breast cancer cell line MCF-7 with inducible expression of dominant negative c-Src. Inactivation of c-Src in these cells inhibited self-renewal and stem cell markers (Nanog, Oct3/4, ALDH1) expression [161]. Furthermore, Tescalcin/c-Src/IGF1Rβ-mediated activation of STAT3 enhances cancer stemness and radioresistance of the A459 lung cancer cell line [162]. Inhibition of connexin43-c-Src signaling with peptide inhibitor reverses glioma stem cell phenotype, suggesting the role of connexin43-c-Src signaling in glioma stem cells [163]. The role of c-Src in cancer stem cells was supported by other studies [164,165]. Mesenchymal and mesenchymal stem cell-like subtypes of triple-negative breast cancer (TNBC), which are more aggressive, lost the expression of miR-34a with subsequent upregulation of its targets. miR-34a restoration or depletion of Src in TNBC cell lines representing these subtypes inhibited proliferation and invasion, activated senescence, and promoted sensitivity to dasatinib, suggesting a role of c-Src in tumor heterogeneity [166].

## 8. Conclusions and Future Directions

Since as far back as the 20th century, a great deal of research has been conducted to understand cancer and the factors that result in carcinogenesis. With the groundbreaking work of Peyton Rous, Hidesaburo Hanafusa, Bishop, and Varmus, a scientific explanation for the causes of cancers was established, particularly the discovery of the Src oncogene. The Src protein signal transduction pathway is used as a tool for cancer cells to evade apoptosis and facilitate cell survival, resulting in poor prognosis [105]. The major pathways activated by c-Src include the Src-JAK-STAT3 pathway, the Src Ras-MAPK/ERK pathway, the Src-PI3K-AKT-mTOR pathway, and the Src-FAK-Paxillin pathway. Scientists are working tirelessly to develop a drug that can effectively target the Src and its partner molecules, hoping to provide more targeted and effective treatments. This has led to the development and approval of Src and multi-kinase inhibitors such as dasatinib, bosutinib, and ponatinib [167]. A comprehensive list of completed and ongoing clinical trials with Src inhibitors is provided in Table 1. Many of the completed early-phase trials demonstrate the safety and efficacy of Srk inhibitors in combination with other drugs. Attention should be paid to details such as schedules, dosages, pharmacological characteristics of each component, availability of biomarkers, etc. during developing new combination therapy or conducting Phase 2/3 trials measuring efficacy.

Molecularly targeted drugs, including small molecule kinase inhibitors and monoclonal antibodies, are quickly changing the landscape of cancer therapy. Survival for almost all cancer types has improved. However, the emergence of resistance is a big hurdle for many patients. It is essential for healthcare professionals to closely monitor patients undergoing chemotherapy and adjust treatment plans as necessary to combat drug resistance [168]. Understanding the molecular mechanism of resistance is critical for developing new combinatorial approaches to combat drug resistance. New research efforts should be directed toward uncovering the precise allosteric sites in Src kinase. By identifying these specific sites, virologists and clinicians can fully grasp the mechanisms underlying the development of resistance, which may lead to novel insights that are needed to develop more effective treatments in the future. Single-cell genomics opens a new avenue to understanding tumor heterogeneity. More such studies will help to understand the role of c-Src in this process and pave the way for developing new combination regimens that are safe and effective. Additionally, a more comprehensive understanding of the various genes and proteins that are involved in the Src signaling pathway can potentially enhance the overall efficacy of chemotherapy treatment in the context of tumorigenesis and resistance.

**Table 1 cancers-16-00032-t001:** Ongoing and completed clinical trials with Src kinase inhibitors.

Type	Drug and Treated Condition	Treatment Regimen	Measuring Outcome(s) and Status
Phase 1b/2a Double-blind, randomized, placebo-controlled trial (NCT04598919)	Saracatinib: SFK inhibitor Condition: Idiopathic pulmonary fibrosis	Oral administration of 125 mg saracatinib and placebo per day for 24 weeks.	Safety and tolerability in patients with Idiopathic Pulmonary Fibrosis Serum level of saracatinib Change in lungs forced vital capacity. Change in serum β-CTX Status: Ongoing
Phase 1 Dose escalation study (NCT05873686)	NXP900: Src/Yes1 inhibitor Condition: Advanced solid tumors	Escalating doses of NXP900 with a starting dose of 20 mg orally per day for 28 days.	Safety and tolerability Pharmacokinetic profile Status: Ongoing
Phase 1 Dose escalation study (NCT00526838)	XL228: Multi-kinase inhibitor including Src Condition: Lymphoma	Dosage 1: 1-h intravenous infusion once a week. Dosage 2: 1-h intravenous infusion twice a week.	Efficacy and pharmacokinetics Safety and tolerability Status: Terminated (sponsor decision)
Phase 1 Dose escalation study (NCT00444015)	Dasatinib: SFK inhibitor Erlotinib: EGFR inhibitor Condition: Non-small cell lung carcinoma (NSCLC)	Patient: 34 Erlotinib tablets starting on Day 1 and dasatinib tablets on Day 9 of a 28-day cycle for 6 cycles. Dose escalation if no dose-limiting toxicities.	Safety and tolerability Determine maximum tolerable dose Pharmacokinetics of erlotinib and dasatinib combination Serum levels of both drugs Serum angiogenic markers Pharmacodynamics Status: Completed Results: The combination of the two drugs are tolerable with disease control and inhibition of plasma angiogenesis markers [169]
Phase 1 Dose escalation study (NCT00658970)	KX2-391: SFK inhibitor Condition: Previously treated advanced solid tumors or lymphoma	Patients: 44 Part 1: Single dose (2, 5, 10 mg) on Day 1 of each 28-day cycle. Part 2: twice daily dosing for 22 days followed by 6 days washout period [170].	Safety and tolerability Pharmacokinetics Status: Completed Results: KX-391 is well tolerated, demonstrates preliminary evidence of biological activity. Recommended maximum tolerated dose is 40 mg BID continuously [170]
Phase 1 Safety Study (NCT00646139)	KX2-391: SFK inhibitor Condition: Previously treated advanced solid tumors or lymphoma	Patients: 7 Oral administration of KX2-391 one or two times per day for 3 weeks.	Safety and tolerability Pharmacokinetics Status: Completed Results: Not available
Phase 2 (NCT00277329)	XL999: a multi-kinase inhibitor of Src, VEGFR, PDGFR Condition: NSCLC	Once weekly, 4-h IV infusion of XL999 at 2.4 mg/kg for 8 weeks.	Safety and subject survival Efficacy and tolerability Pharmacokinetic and pharmacodynamic profile of XL999 Status: Terminated due to cardiotoxicity of XL999
Phase 2a (NCT02167256)	AZD0530 (Saracatinib)—SFK inhibitor Condition: Mild Alzheimer zczc disease.	Patients: 159 50% of the subjects were placed on 100 mg of AZD0530 daily. Patients with a plasma drug level of less than 100 ng/mL after two weeks received 125 mg of AZD0530 daily, while the other 50% received a placebo.	Adverse effects. Change in brain glucose uptake. Change in cognitive and behavioral functions. Status: Completed Results: No significant improvement [171]
Phase 2 Randomized double-blind study (NCT00752206)	Saracatinib: SFK inhibitor Condition: Recurrent osteosarcoma localized to the lungs.	Patients: 38 (37 analyzed) 175 mg of saracatinib or placebo, orally for a 28-day cycle for 13 cycles.	Disease progression. Y-year overall survival Genes associated with osteosarcoma and genetic mutations. Activation of Src kinase Status: Terminated with results Results: No improvement, Src inhibition alone may not be sufficient to suppress metastatic progression [172]
Phase 1 (NCT01482728)	Dasatinib-SFK inhibitor Condition: Endometrial cancer	Patients: 12 (10 completed) 100 or 200 mg dasatinib the day before surgery and the day of surgery.	Changes in levels of SFK protein activity in endometrial tumor tissue And blood Status: Completed Results: All patients had reduction in at least one Src parameter in either tissue or blood [173]
Phase I Biomarker comparison (NCT00779389)	Dasatinib-SFK inhibitor Erlotinib: EGFR inhibitor Condition: Head and neck cancer; NSCLC	Patients: 58 Arm A: Erlotinib 150 mg once a day for 14–21 days. Arm B: Dasatinib (100 mg) + Placebo, once a day for 14–21 days. Arm 3: Erlotinib (150 mg) + dasatinib (100 mg), PO qD for 14–21 days.	Biomarker modulation Status: Completed Results: Significant decrease in tumor size in both erlotinib arms, no effect was seen with dasatinib alone [174]
Phase 1 (NCT01999985)	Afatinib: EGFR inhibitor Dasatinib: SFK inhibitor Condition: NSCLC	Patients: 25 Dasatinib 1A: Begins Day 8. Level 1–100 mg; Level 2–100 mg; Level 3: 140 mg. Afatinib 1A: Begins Day 1. Level 1–30 mg; Level 2–40 mg; Level 3–40 mg. Dasatinib and Afatinib1B: recommended dose from 1A.	Safety and tolerability Response of participants and progression-free survival. Status: Completed Results: Manageable toxicity profile; modulation of T790M mutation; no objective clinical response [175]
Phase 1 (NCT00672295)	Dasatinib: SFK inhibitor Paclitaxel: Micritibule inhibitor Carboplatin: DNA alkylating agent Condition: Ovarian cancer, peritoneal cancer, fallopian tube cancer	Patients: 20 Dasatinib: 50–250 mg, every day on Days 2–21 in the first cycle (3 weeks) and continuously (Days 1–21) throughout the remainder of the therapy. Paclitaxel: 150–175 mg/m^2^, IV infused over 3 h on Day 1 of each cycle. Carboplatin (AUC = 5–6 mg, I/min), IV infused over 30–60 min on Day 1 of each cycle.	Pharmacokinetic and pharmacodynamic parameters of dasatinib, paclitaxel, and carboplatin. Toxicity of the combination of the three drugs in the subjects. Status: Completed Result: Due to the high incidence of myelosuppression, the recommended Phase 2 dose of dasatinib is 150 mg daily in combination with paclitaxel and carboplatin [176]
Phase 2 Neoadjuvant study (NCT01990196)	Dasatinib: SFK inhibitor Degarelix + enzalutamide: Androgen receptor (AR) inhibitors Trametinib: MEK inhibitor Condition: Prostate cancer	Group 1 (AR inhibition only): 240 mg degarelix SQ as a starting dose, followed by 80 mg every 4 weeks. 160 mg enzalutamide orally once daily. Group 2 (AR inhibition + MEK inhibition): Four weeks after androgen inhibition, 2 mg of trametinib orally for two to four weeks. Group 3 (AR inhibition + Src inhibition): Four weeks after androgen inhibition, 100 mg dasatinib oral daily.	Expression of N-cadherin and vimentin Status: Ongoing
Phase 1 (NCT00501410)	Dasatinib: SFK inhibitor Cetuximab: EGF inhibitor FOLFOX (5-FU + Leucovorin + Oxaliplatin) 5-FU: Thymidylate synthase inhibitor Oxaliplatin: DNA alkylating agent Leucovorin: Folate analog Condition: Metastatic colorectal cancer	Patients: 77 Oral dasatinib 100 mg from Day 1–14. IV cetuximab 400 mg/m^2^ followed by 250 mg/m^2^ weekly on Days 1 and 8. IV 5-FU 2400 mg/m^2^ on Days 1 and 2. IV leucovorin 400 mg/m^2^ on Day 1. IV oxaliplatin 85 mg/m^2^ on Day 1.	Safety and tolerability of the drugs. Dasatinib response rate with modified FOLFOX therapy. Status: Completed Result: The combination of dasatinib plus FOLFOX ± cetuximab showed modest clinical activity. Incomplete inhibition of Src at the dose level [177]
Phase 1 (NCT01668550)	AZD0424: Src/Abl inhibitor Condition: Advanced solid tumor	Patients: 43 Daily oral administration of AZD040.	Safety and tolerability. Status: Terminated due to lack of preclinical efficacy Result: Not available
Phase 1 (NCT01015222)	Dasatinib: SFK inhibitor Bevacizumab: VEGF inhibitor Paclitaxel: Microtubule inhibitor Methylnaltrexone Condition: Advanced cancer	Patients: 122 Dasatinib: 50 mg every day for 28 days, followed by dose escalation. Bevacizumab: IV 5 mg/kg on Days 1 and 15, followed by dose escalation. Paclitaxel: IV 40 mg/m^2^ on Days 1, 8, and 15, followed by dose escalation.	Safety and tolerability Efficacy of the combination of the three drugs with or without methylnaltrexone. Status: Completed Result: Not available
Phase 1 (NCT01445509)	Dasatinib: SFK inhibitor Bevacizumab: VEGF inhibitor Condition: Advanced solid tumors	Patients: 50 Arm1: Oral dasatinib (50 mg) and IV bevacizumab (5 mg/kg) simultaneously every two weeks within 28-day treatment cycles. The dosage is escalated in cohorts of three to six patients until the most suitable and safe dose is identified. Arm2: Patients are randomly allocated to receive either dasatinib or bevacizumab during the first treatment cycle, followed by both drugs in all subsequent treatment cycles.	Safety and tolerability Biochemical levels of Src-FAK, Src-PLC, and VEGF Status: Completed Result: Not published
Phase 2 (NCT00528645) Single group assignment trial	AZD0530—SFK inhibitor Condition: Extensive stage small cell lung cancer	Patients: 23 175 mg/day orally for 2 years in the absence of disease progression or unacceptable toxicity.	Progression-free survival Tumor response Status: Completed Result: Tolerable at 175 mg/day dose without improvement in progression-free survival [178]
Phase 1/2 (NCT03041701)	Dasatinib: SFK inhibitor Ganitumab: IGF-1R inhibitor Condition: Embryonal and alveolar rhabdomyosarcoma	Once daily oral administration of dasatinib on Days 7–27 during cycle 1 and then Days 0–27 for subsequent cycles. Once every 2 weeks beginning on Day 0.	Safety and tolerability Progression-free survival Clinical response to pharmacotherapy Status: Terminated without result (Unavailability of drug)
Phase 1/2 (NCT01306942)	Dasatinib: SFK inhibitor Trastuzumab: Human epidermal growth factor receptor 2 (Her2) inhibitor Paclitaxel: Microtubule inhibitor Condition: Her2-positive metastatic breast cancer	Patients: 37 IV loading dose of 4 mg/kg of trastuzumab in cycle 1 followed by 2 mg/kg in every cycle. 80 mg/m^2^ weekly of paclitaxel. Oral two-level doses (100 and 140 mg) of dasatinib once daily.	Safety and tolerability Efficacy and clinical benefit rate Pharmacokinetics, time to progression, progression-free survival, and response duration. Status: Completed Result: Phase 1 part suggests the feasibility of the combination of dasatinib, trastuzumab, and paclitaxel [179]; Phase 2 results are not available.
Phase 2 (NCT00780676)	Dasatinib: SFK inhibitor AZD6244 (Selumetinib): MEK inhibitor Condition: Metastatic breast cancer	Patients: 97 Oral administration of 100 mg of dasatinib daily. 75 mg of selumetinib administered orally twice daily.	Clinical benefit Status: Terminated (Lack of significant result)
Phase 1 (NCT00996723)	Dasatinib: SFK inhibitor Vandetanib: Src, VEGFR2, EGFR, and Rearranged during transfection (RET) inhibitor. Condition: Diffuse Intrinsic Pontine Glioma	Patients: 25 Oral administration of both drugs during and after local radiation therapy.	Safety and tolerability Pharmacokinetics and pharmacodynamics of both drugs. Influence of CYP3A4/5 Status: Completed Result: The maximum tolerable dose of vandetanib and dasatinib in combination is 65 mg/m^2^ for each drug [180]
Phase 1	Dasatinib: SFK inhibitor Crizotinib: Multi-kinase inhibitor Condition: Diffuse intrinsic pontine glioma	Dasatinib: 50 mg/m^2^ Crizotinib: 100, 130 mg/m^2^ TID, 215 mg/m^2^ once daily.	Safety and tolerability Status: CompletedResults: This drug combination is poorly tolerable [181]

## Figures and Tables

**Figure 1 cancers-16-00032-f001:**
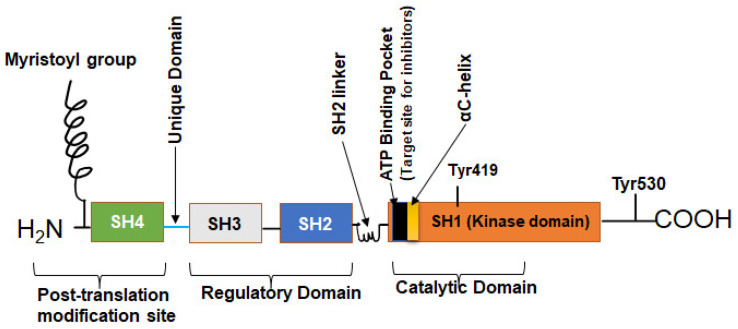
The structure of the human c-Src. The human Src protein is an orderly arrangement of four SH domains with N and C terminals. The C-terminal is the tail and anchors a regulatory tyrosine—Tyr530. SH1 domain is the highly catalytic tyrosine kinase (TK) domain with Tyr419 for substrate phosphorylation, ATP/inhibitor binding pocket, and an αC-helix. SH2 and SH3 domains are the regulatory domains that can bind phosphorylated Tyr530 and polyproline helix, respectively. The SH4 domain is the disordered segment and myristoylation site for membrane association [24,25].

**Figure 2 cancers-16-00032-f002:**
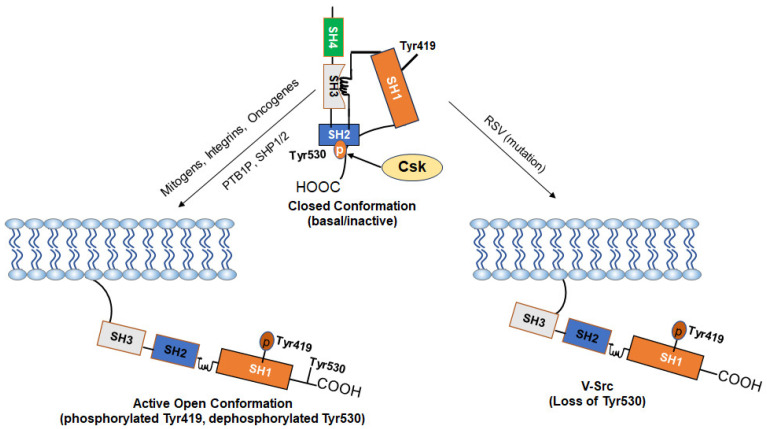
Mechanism of Activation of c-Src: The kinase activity of c-Src is tightly regulated by the phosphorylation and dephosphorylation of Tyr530 and Tyr519 and depends on whether the kinase is in a “closed” or an “open” conformation. In normal cells, the basal activity of c-Src is maintained via a negative regulation by Tyr530. Phosphorylation of Tyr530 by the C-terminal Src kinase (Csk) enables high-affinity binding of phospho-Tyr530 to the SH2 domain, leading to the more compact “closed” conformation (inactive form). The “closed” conformation is further stabilized through interactions between the SH3 domain and a proline-rich region present in the kinase domain. Stimuli such as growth factors and integrins cause dephosphorylation of Tyr530 via activation of phosphatases (PTB1B, SHP1/2) or inactivation of Csk and disrupt the interaction of the SH2 domain with phospho-Tyr530, leading to an open conformation and allow autophosphorylation of Tyr419 (active form). Loss of the negative-regulatory C-terminal region due to viral integration in v-Src makes it CA as an oncogene. Some driver oncogenes constitutively activate c-Src.

**Figure 3 cancers-16-00032-f003:**
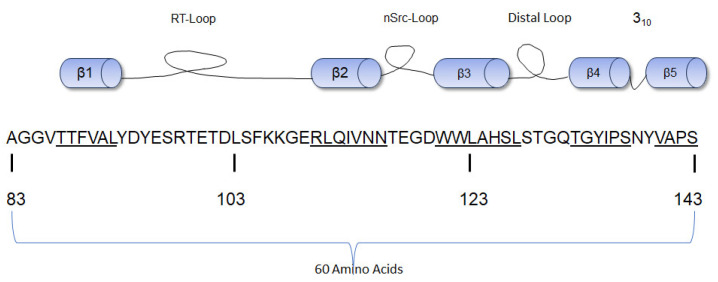
Diagrammatic representation of the β-strands of the SH3 domain of human c-Src with the amino acid numbering [54]. The sequence of each β strand is underlined.

**Figure 4 cancers-16-00032-f004:**
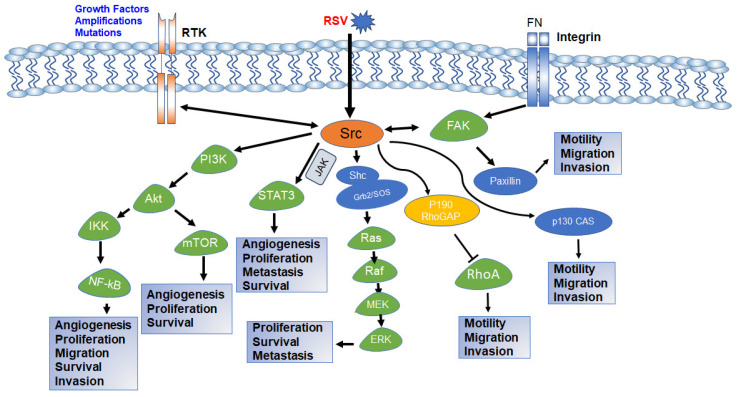
Schematic diagram of Src intracellular signaling pathway and various cellular processes controlled by each pathway. Activation of RTK, integrin signaling, or viral integration activates c-Src. Activated c-Src activates downstream PI3K-AKT, STAT3, Ras-MAPK, and FAK pathways and regulates various cellular functions required for normal growth and development as well as carcinogenesis and drug resistance. The both-way arrow indicates that the components can be activated by each other [88,89,90,91,92].

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
