# Peer review of "Role of c-Src in Carcinogenesis and Drug Resistance"

_cancers, 2023, doi:10.3390/cancers16010032_

Round 1

Reviewer 1 Report

Comments and Suggestions for Authors

The manuscript entitled “Role of c-Src in Carcinogenesis and Drug Resistance” focuses on the description of the role of c-Src gene dysregulation in cancer development through participation of c-Src protein in various signaling pathways to promote oncogenesis and resistance to chemotherapy drugs. This paper scientifically sounds and may be of interest to the journal audience. The authors provided very interesting history of c-Scr and its structure and functioning. However, I have some concerns which should be addressed before publication:

1.     In the Abstract, I recommend to indicate signaling pathways discussed in the manuscript.

2.     Graphical Abstract – cancer cells at different stages of cancer progression are shown as the same manner. It is recommended to show that cells are changed (morphologically, functionally etc.).

3.     Section “SH3 domain”, paragraph 1, sentences 2 and 3, lines 268-270 – it is not unclear, which two peptides the authors are telling about? What are the class I and II peptides? Figure 3, how many amino acid residues is each beta-strand composed of? Also, on the figure, loops between beta-strands look like alpha-helices, but not loops. Also, the sentence on lines 280-282 – it is unclear, which amino acid residues is a lipid-binding region composed of?

4.     Why did the authors not provide figures to illustrate structures of other domains? It is also recommended to provide a figure on c-Src activation?

5.     Figure 4, captions given in blue-brown boxes are poorly readable. It is recommended to change the color of these boxes.

6.     There have been other review articles published last years on Src functions. What new contribution your manuscript will make in the field?   

7.     Not all abbreviations are explained at their first usage.

8.     Numerations of sections and sub-sections are inappropriate. This should be checked and corrected.

Comments on the Quality of English Language

The English language is quite well and readable.

Author Response

A point-by-point response to reviewers’ comments

We are thankful to the editor and the reviewers for their enthusiasm, careful reading of the manuscript, positive impressions, and pinpointing the weaknesses. Our response to your concerns is given below.  

Thank you so much for your constructive comments. Our point-by-point responses are given below.

  1. In the Abstract, I recommend to indicate signaling pathways discussed in the manuscript.

Response: As per the suggestion, we included these pathways in the abstract of the revised manuscript (Page 1, lines 27-30).

  1. Graphical Abstract – cancer cells at different stages of cancer progression are shown as the same manner. It is recommended to show that cells are changed (morphologically, functionally, etc.).

Response: As per the recommendation, we have modified the graphical abstract to include different morphology for different stages of cancer (Page 2, revised manuscript).

  1. Section “SH3 domain”, paragraph 1, sentences 2 and 3, lines 268-270 – it is not unclear, which two peptides the authors are telling about? What are the class I and II peptides? Figure 3, how many amino acid residues is each beta-strand composed of? Also, on the figure, loops between beta-strands look like alpha-helices, but not loops. Also, the sentence on lines 280-282 – it is unclear, which amino acid residues is a lipid-binding region composed of?

Response: We have rewritten this section to clarify Class I and II peptides (Page 6, lines 285-290). We have also redrawn Figure 3 with the sequence of each β strand underlined. We have included the amino acid residue numbers in the revised manuscript (highlighted in lines 299 and 300 in the revised version).

  1. Why did the authors not provide figures to illustrate structures of other domains? It is also recommended to provide a figure on c-Src activation?

Response: We have provided the detailed structure of the SH3 domain because of its complexity. The important residues in other domains are denoted in Figures 1 and 2. The purpose of Figure 2 was to describe the activation of c-Src. We have revised Figure 2 and the legend for Figure 2 to describe the mechanism of activation of c-Src (page 5 in the revised manuscript).

  1. Figure 4, captions given in blue-brown boxes are poorly readable. It is recommended to change the color of these boxes.

Response: As per the recommendation, we have revised Figure 4 to make it simple but relate to the major messages.

  1. There have been other review articles published last years on Src functions. What new contribution your manuscript will make in the field?   

Response: We agree with the reviewer that many review articles are published each year describing c-Src. We believe that ours is a comprehensive review that provides a detailed history of the discovery of c-Src as the first identified protooncogene, its structure, normal physiological functions in addition to its role in carcinogenesis and drug resistance. We expect that the article will be well-accepted by the readers.

  1. Not all abbreviations are explained at their first usage.

Response: We were much more careful about the abbreviations during the revision and hope that each abbreviation is expanded during its first appearance.

  1. Numerations of sections and sub-sections are inappropriate. This should be checked and corrected.

Response: Thanks for bringing this mistake to our attention. We have corrected the section numbering.

Reviewer 2 Report

Comments and Suggestions for Authors

Overall this is a really well done review of Src kinase. Even though much is known about Src kinase this review is relevant because of what we continue to learn regarding how Src enhances cancer progression and drug resistance, which has profound implications on human health. The historical perspective does a nice job of setting the stage for the overview of how Src interacts with various cell signaling pathways.

Major

My main criticism of the paper is that, while the authors do a nice job of describing Src kinase structure and function, they don’t really begin to address Src function in carcinogenesis until ¾ of the way through the review. And the drug resistance section toward the end is fairly brief. Outside of giving some brief descriptions they don't go into much depth regarding the resistance issue. The manuscript would better match the title if the authors were to expand the Src carcinogenesis and drug resistance sections.

Secondary criticism is that the figures are not very easy to read. More effort could be out into making them more visually easier to interpret.

Minor

Line 186 Michael is misspelled as Micheal

Line 199 myristoylation is misspelled as myristylation (they use myristoyl group in Fig 11) 

Fig 2 – hard to see/read with the small domain labels, etc., what happened to the SH4 domain? It’s not clear how it’s being represented.

Line 352 lungs should be lung (not plural)

Fig 4 – too hard to read – there is so much going on that it’s hard to tell what the authors want the reader to understand from the figure

              The arrows are to chaotic – very hard to follow

              The red “t” inhibition line between ERK and ERK inhibitors seems to be flipped in the wrong direction

              The small black text in the blue/orange boxes is too hard to read – need to pick another color combination lighter background with the black font or lighter font with the current background

Lines 512 – 518 font size changed

*the authors also use the phrase "lipidic" with respect to membranes. This is an awkward use of that phrasing, which is not incorrect, but most literature uses lipid membrane as the common phrasing.

Author Response

Thank you so much for your encouraging comments and constructive criticisms. We have followed your recommendations as much as possible and believe that the manuscript has been greatly improved. Our point-by-point response to your comments is appended below.

Major concern 1: My main criticism of the paper is that, while the authors do a nice job of describing Src kinase structure and function, --------------------. The manuscript would better match the title if the authors were to expand the Src carcinogenesis and drug resistance sections.

Response: We fully agree with the reviewer that a major part of the manuscript describes the history of discovery and structural features of c-Src. Although the title focuses on the role of c-Src in carcinogenesis and drug resistance, we have tried to write a comprehensive review of c-Src so that readers find all the necessary information including the structure and normal physiological roles in this review. We strongly believe that the majority of the audience will like it and be benefited.

Major concern 2: Secondary criticism is that the figures are not very easy to read. More effort could be out into making them more visually easier to interpret.

Response: We have redrawn all figures to make them simple but more informative and easy to interpret.

Minor concerns

Line 186 Michael is misspelled as Micheal

Response: We have corrected the spelling (line180 in the revised manuscript)

Line 199 myristoylation is misspelled as myristylation (they use myristoyl group in Fig 11) 

Response: Corrected (line 214 in the revised manuscript)

Fig 2 – hard to see/read with the small domain labels, etc., what happened to the SH4 domain? It’s not clear how it’s being represented.

Response: The figure has been re-drawn to make it easily readable with the SH4 domain

Line 352 lungs should be lung (not plural)

Response: Corrected (line 409)

Fig 4 issues

Response: We have re-drawn Figure 4 to make it simple and easily readable. Thanks for raising the issues.

Lines 512 – 518 font size changed

Response: Corrected (lines 586-594)

*the authors also use the phrase "lipidic" with respect to membranes. This is an awkward use of that phrasing, which is not incorrect, but most literature uses lipid membrane as the common phrasing.

Response: Rephrased to make it more acceptable.

Reviewer 3 Report

Comments and Suggestions for Authors

The authors conduct a comprehensive review of c-Src, a proto-oncogene that plays a major role in carcinogenesis and drug resistance in cancer. It traces the history of c-Src's discovery, from Peyton Rous' early experiments showing the transmission of tumors in chickens to the later identification of the cellular Src (c-Src) gene by Bishop and Varmus. The article then delves into the structure and activation mechanisms of c-Src, describing its multiple domains (SH1, SH2, SH3, SH4) and how signals from growth factors, receptors, and binding partners allow tight regulation under normal conditions but can lead to overexpression in cancerous states. c-Src activates downstream pathways like JAK/STAT, MAPK/ERK, PI3K/AKT, and FAK/paxillin that control proliferation, survival, invasion and other cancer hallmarks.

The article also focuses on how elevated c-Src facilitates drug resistance through these same pathways, reducing chemotherapy efficacy and posing a major clinical challenge. Various cancer types are impacted, including breast, ovarian, colon, lung and head/neck cancers. However, new small molecule Src inhibitors show promise to reverse resistance, especially when given in combination with other drugs. Future directions center on fully elucidating c-Src regulation mechanisms and drug synergy approaches to improve outcomes for cancer patients.

  1. The article could state specific cancer cell lines (e.g. MCF7, A549) and animal models (xenograft, genetic) utilized in referenced studies linking c-Src to cancer outcomes and drug resistance. Details on origin, genetic status and relevant markers would allow better evaluation.
  2. A section reviewing c-Src's normal role in processes like cellular growth, division, adhesion, migration could put pathological activation in better context. Comparisons of signaling intensity/duration between normal physiology and disease state could be informative.
  3. For key statements about c-Src enabling particular hallmarks of cancer, the underlying molecular mechanisms could be explained in more detail (e.g. c-Src phosphorylation of which residues on a protein activates that protein's capacity to drive proliferation). Diagrams mapping specific cascades inside cells could make interactions and downstream effects clearer.
  4. Discussing completed or ongoing clinical trials exploring Src inhibitors in cancer patients (compound, phase, cancers, combinatorial drugs, outcomes) would reveal translational landscape and challenges that remain in targeting this pathway.
  5. To support verbal descriptions of c-Src activations and effects, quantitative diagrams like dose-response curves, kinetic activation plots, signaling network diagrams could better depict key variables and relationships.
  6. Qualifying statements about c-Src's involvement in processes with phrases like "may contribute partially to", "likely collaborates with X", "enhances but alone doesn't drive" would enhance accuracy and avoid overstating case.
  7. Optimizing synergistic combinations requires studying details like schedules and dosages. Research on factors like overlapping toxicities and biomarker-directed alternating regimens could provide practical clinical insights on translating synergy potential with c-Src inhibitors.
  8. Expanding discussion to c-Src's role in tumor heterogeneity and cancer stem cell maintenance could offer additional angles for therapy development, like targeting resistance mechanisms or preventing recurrence.

Author Response

  1. The article could state specific cancer cell lines (e.g. MCF7, A549) and animal models (xenograft, genetic) utilized in referenced studies linking c-Src to cancer outcomes and drug resistance. Details on origin, genetic status and relevant markers would allow better evaluation.

Response: We have included this information in the revised manuscript as much as possible.

  1. A section reviewing c-Src's normal role in processes like cellular growth, division, adhesion, migration could put pathological activation in better context. Comparisons of signaling intensity/duration between normal physiology and disease state could be informative.

Response: Thanks for this suggestion. We have included the normal physiological function of c-Src in section 4 (Page 7-8).

  1. For key statements about c-Src enabling particular hallmarks of cancer, the underlying molecular mechanisms could be explained in more detail (e.g. c-Src phosphorylation of which residues on a protein activates that protein's capacity to drive proliferation). Diagrams mapping specific cascades inside cells could make interactions and downstream effects clearer.

Response: We tried our best to provide as detailed information as possible from each reference.

  1. Discussing completed or ongoing clinical trials exploring Src inhibitors in cancer patients (compound, phase, cancers, combinatorial drugs, outcomes) would reveal translational landscape and challenges that remain in targeting this pathway.

Response: Again, thank you so much for this suggestion. We have included a table (Table 1) to include clinical trials that involve c-Src inhibitors.

  1. To support verbal descriptions of c-Src activations and effects, quantitative diagrams like dose-response curves, kinetic activation plots, signaling network diagrams could better depict key variables and relationships.

Response: A comprehensive network diagram is provided in Figure 4 along with available inhibitors for each target in the network.

  1. Qualifying statements about c-Src's involvement in processes with phrases like "may contribute partially to", "likely collaborates with X", "enhances but alone doesn't drive" would enhance accuracy and avoid overstating case.

Response: We appreciate the reviewer’s suggestion. We carefully read each reference to avoid overstatement as much as possible.

  1. Optimizing synergistic combinations requires studying details like schedules and dosages. Research on factors like overlapping toxicities and biomarker-directed alternating regimens could provide practical clinical insights on translating synergy potential with c-Src inhibitors.

Response: We include this discussion in the future direction.

  1. Expanding discussion to c-Src's role in tumor heterogeneity and cancer stem cell maintenance could offer additional angles for therapy development, like targeting resistance mechanisms or preventing recurrence.

Response: This is a great suggestion. In the revised manuscript, we have written a new section (section 7) to discuss the role of c-Src in cancer stem cells and heterogeneity.

Round 2

Reviewer 1 Report

Comments and Suggestions for Authors

All my recommendations have been properly addressed

Comments on the Quality of English Language

English language is good

Reviewer 2 Report

Comments and Suggestions for Authors

Appreciate the effort to revise figures so they are more easily interpreted.

Reviewer 3 Report

Comments and Suggestions for Authors

All of my comments have addressed